

# A neural network approach to estimate a posteriori distributions of Bayesian retrieval problems

Simon Pfreundschuh[1], Patrick Eriksson[1], David Duncan[1], Bengt Rydberg[2], Nina Håkansson[3], and Anke Thoss[3]

[1]Department of Space, Earth and Environment, Chalmers University of Technology, Gothenburg, Sweden
[2]Möller Data Workflow Systems AB, Gothenburg, Sweden
[3]Swedish Meteorological and Hydrological Institute (SMHI), Norrköping, Sweden

**Correspondence:** simon.pfreundschuh@chalmers.se

**Abstract.**

This work is concerned with the retrieval of physical quantities from remote sensing measurements. A neural network based method, Quantile Regression Neural Networks (QRNNs), is proposed as a novel approach to estimate the a posteriori distribution of Bayesian remote sensing retrievals. The advantage of QRNNs over conventional neural network retrievals is that they not only learn to predict a single retrieval value but also the associated, case specific uncertainties. In this study, the retrieval performance of QRNNs is characterized and compared to that of other state-of-the-art retrieval methods.

A synthetic retrieval scenario is presented and used as a validation case for the application of QRNNs to Bayesian retrieval problems. The QRNN retrieval performance is evaluated against Markov chain Monte Carlo simulation and another Bayesian method based on Monte Carlo integration over a retrieval database. The scenario is also used to investigate how different hyperparameter configurations and training set sizes affect the retrieval performance. In the second part of the study, QRNNs are applied to the retrieval of cloud top pressure from observations by the moderate resolution imaging spectroradiometer (MODIS). It is shown that QRNNs are not only capable of achieving similar accuracy as standard neural network retrievals, but also provide statistically consistent uncertainty estimates for non-Gaussian retrieval errors.

The results presented in this work show that QRNNs are able to combine the flexibility and computational efficiency of the machine learning approach with the theoretically sound handling of uncertainties of the Bayesian framework. Together with this article, a Python implementation of QRNNs is released through a public repository to make the method available to the scientific community.

## 1 Introduction

The retrieval of atmospheric quantities from remote sensing measurements constitutes an inverse problem that generally does not admit a unique, exact solution. Measurement and modeling errors, as well as limited sensitivity of the observation system, preclude the assignment of a single, discrete solution to a given observation. A meaningful retrieval should thus consist of a retrieved value and an estimate of uncertainty describing a range of values that are likely to produce a measurement similar to





the one observed. However, even if a retrieval method allows for explicit modeling of retrieval uncertainties, their computation and representation is often possible only in an approximate manner.

The Bayesian framework provides a formal way of handling the ill-posedness of the retrieval problem and its associated uncertainties. In the Bayesian formulation (Rodgers, 2000), the solution of the inverse problem is given by the a posteriori distribution $p(x|\boldsymbol{y})$, i.e. the conditional distribution of the retrieval quantity $x$ given the observation $\boldsymbol{y}$. Under the modeling assumptions, the posterior distribution represents all available knowledge about the retrieval quantity $x$ after the measurement, accounting for all considered retrieval uncertainties. Bayes' theorem states that the a posteriori distribution is proportional to the product $p(\boldsymbol{y}|x)p(x)$ of the a priori distribution $p(x)$ and the conditional probability of the observed measurement $p(\boldsymbol{y}|x)$. The a priori distribution $p(x)$ represents knowledge about the quantity $x$ that is available before the measurement and can be used to aid the retrieval with supplementary information.

For a given retrieval, the a posteriori distribution can generally not be expressed in closed form and different methods have been developed to compute approximations to it. In cases that allow a sufficiently precise and efficient simulation of the measurement, a forward model can be used to guide the solution of the inverse problem. If such a forward model is available, the most general technique to compute the a posteriori distribution is Markov chain Monte Carlo (MCMC) simulation. MCMC denotes a set of methods that iteratively generate a sequence of samples, whose sampling distribution approximates the true a posteriori distribution. MCMC simulations have the advantage of allowing the estimation of the a posteriori distribution without requiring any simplifying assumptions on a priori knowledge, measurement error or the forward model. The disadvantage of MCMC simulation is that each retrieval requires a high number of forward model evaluations, which in many cases makes the method computationally too demanding to be practical. For remote sensing retrievals, the method is therefore of interest rather for testing and validation (Tamminen and Kyrölä, 2001), such as for example in the retrieval algorithm developed by Evans et al. (2012).

A method that avoids costly forward model evaluations during the retrieval has been proposed by Kummerow et al. (1996). The method is based on Monte Carlo integration of importance weighted samples in a retrieval database $\{(\boldsymbol{y}_i, x_i)\}_{i=0}^{n}$, which consists of pairs of observations $\boldsymbol{y}_i$ and corresponding values $x_i$ of the retrieval quantity. The method will be referred to in the following as Bayesian Monte Carlo integration (BMCI). Even though the method is less computationally demanding than methods involving forward model calculations during the retrieval, it may require the traversal of a potentially large retrieval database. Furthermore, the incorporation of ancillary data to aid the retrieval requires careful stratification of the retrieval database, as it is performed in the Goddard Profiling Algorithm (Kummerow et al., 2015) for the retrieval of precipitation profiles. Further applications of the method can be found for example in the work by Rydberg et al. (2009) or Evans et al. (2012).

The optimal estimation method (Rodgers, 2000), in short OEM (also MAP, 1DVAR), simplifies the Bayesian retrieval problem assuming that a priori knowledge and measurement uncertainty both follow Gaussian distributions and that the forward model is only moderately non-linear. Under these assumptions, the a posteriori distribution is approximately Gaussian. The retrieved values in this case are the mean and maximum of the a posteriori distribution, which coincide for a Gaussian distribution, together with the covariance matrix describing the width of the a posteriori distribution. In cases where an efficient



forward model for the computation of simulated measurements and corresponding Jacobians is available, the OEM has become the quasi-standard method for Bayesian retrievals. Nonetheless, even neglecting the validity of the assumptions of Gaussianity and linearity, the method is unsuitable for retrievals that involve complex radiative processes. In particular, since the OEM requires the computation of the Jacobian of the forward model, complex processes such as surface or cloud scattering become
too expensive to model online during the retrieval.

Compared to the Bayesian retrieval methods discussed above, machine learning provides a more flexible approach to learn computationally efficient retrieval mappings directly from data. Large amounts of data available from simulations, collocated observations or in situ measurements, as well as increasing computational power to speed up the training, have made machine learning techniques an attractive alternative to approaches based on (Bayesian) inverse modeling. Numerous applications of
machine learning regression methods to retrieval problems can be found in recent literature (Jiménez et al., 2003; Holl et al., 2014; Strandgren et al., 2017; Wang et al., 2017; Håkansson et al., 2018; Brath et al., 2018). All of these examples, however, neglect the probabilistic character of the inverse problem and provide only a scalar estimate of the retrieval. Uncertainty estimates in these retrievals are provided in the form of mean errors computed on independent test data, which is a clear drawback compared to Bayesian methods. A notable exception is the work by Aires et al. (2004), which applies the Bayesian framework
to estimate errors in the retrieved quantities due to uncertainties on the learned neural network parameters. However, the only difference to the approaches listed above is that the retrieval errors, estimated from the error covariance matrix observed on the training data, are corrected for uncertainties in the network parameters. With respect to the intrinsic retrieval uncertainties, the approach is thus afflicted with the same limitations. Furthermore, the complexity of the required numerical operations make it suitable only for small training sets and simple networks.

In this article, quantile regression neural networks (QRNNs) are proposed as a method to use neural networks to estimate the a posteriori distribution of remote sensing retrievals. Originally proposed by Koenker and Bassett Jr (1978), quantile regression is a method for fitting statistical models to quantile functions of conditional probability distributions. Applications of quantile regression using neural networks (Cannon, 2011) and other machine learning methods (Meinshausen, 2006) exist, but to the best knowledge of the authors this is the first application of QRNNS to remote sensing retrievals. The aim of this work is to
combine the flexibility and computational efficiency of the machine learning approach with the theoretically sound handling of uncertainties in the Bayesian framework.

A formal description of QRNNs and the retrieval methods against which they will be evaluated are provided in Sect. 2. A simulated retrieval scenario is used to validate the approach against BMCI and MCMC in Sect. 3. Section 4 presents the application of QRNNs to the retrieval of cloud top pressure and associated uncertainties from satellite observations in the
visible and infrared. Finally, the conclusions from this work are presented in Sect. 5.

## 2   Methods

This section introduces the Bayesian retrieval formulation and the retrieval methods used in the subsequent experiments. Two Bayesian methods, Markov chain Monte Carlo simulation and Bayesian Monte Carlo integration, are presented. Quantile



regression neural networks are introduced as a machine learning approach to estimate the a posteriori distribution of Bayesian retrieval problems. The section closes with a discussion of the statistical metrics that are used to compare the methods.

## 2.1 The Retrieval Problem

The general problem considered here is the retrieval of a scalar quantity $x \in \mathrm{R}$ from an indirect measurement given in the
form of an observation vector $\boldsymbol{y} \in \mathrm{R}^m$. In the Bayesian framework, the retrieval problem is formulated as finding the posterior distribution $p(x|\boldsymbol{y})$ of the quantity $x$ given the measurement $\boldsymbol{y}$. Formally, this solution can be obtained by application of Bayes theorem:

$$p(x|\boldsymbol{y}) = \frac{p(\boldsymbol{y}|x)p(x)}{\int p(x', \boldsymbol{y})dx'}. \tag{1}$$

The a priori distribution $p(x)$ represents the knowledge about the quantity $x$ that is available prior to the measurement. The
a priori knowledge introduced into the retrieval formulation regularizes the ill-posed inverse problem and ensures that the retrieval solution is physically meaningful. The a posteriori distribution of a scalar retrieval quantity $x$ can be represented by the corresponding cumulative distribution function (CDF) $F_{x|\boldsymbol{y}}(x)$, which is defined as

$$F_{x|\boldsymbol{y}}(x) = \int\limits_{-\infty}^{x} p(x'|\boldsymbol{y}) \, dx'. \tag{2}$$

## 2.2 Bayesian Retrieval Methods

Since the a posteriori distribution in Eq. (1) can generally not be computed or sampled from directly, numerous methods were developed to approximate the a posteriori distribution to varying degrees of accuracy.

### 2.2.1 Markov chain Monte Carlo

Markov chain Monte Carlo (MCMC) simulation denotes a set of methods for the generation of samples from arbitrary posterior distributions $p(x|\boldsymbol{y})$. The general principle is to compute samples from an approximate distribution and refine them in a way
such that their distribution converges to the true a posteriori distribution (Gelman et al., 2013). In this study, the Metropolis algorithm is used to implement MCMC. The Metropolis algorithm iteratively generates a sequence of states $\boldsymbol{x}_0, \boldsymbol{x}_1, \ldots$ using a symmetric proposal distribution $J_t(\boldsymbol{x}^*|\boldsymbol{x}_{t-1})$. In each step of the algorithm, a proposal $\boldsymbol{x}^*$ for the next step is generated by sampling from $J_t(\boldsymbol{x}^*|\boldsymbol{x}_{t-1})$. The proposed state $\boldsymbol{x}^*$ is accepted as the next simulation step $\boldsymbol{x}_t$ with probability $\min\left\{1, \frac{p(\boldsymbol{x}^*|\boldsymbol{y})}{p(\boldsymbol{x}_{t-1}|\boldsymbol{y})}\right\}$. Otherwise $\boldsymbol{x}^*$ is rejected and the current simulation step $\boldsymbol{x}_{t-1}$ is kept for $\boldsymbol{x}_t$. If the proposal distribution $J(\boldsymbol{x}^*, \boldsymbol{x}_{t-1})$ is sym-
metric and samples generated from it satisfy the Markov chain property with a unique stationary distribution, the Metropolis algorithm is guaranteed to produce a distribution of samples which converges to the true a posteriori distribution.





### 2.2.2 Bayesian Monte Carlo integration

The BMCI method is based on the use of importance sampling to approximate integrals over the a posteriori distribution of a given retrieval case. Consider an integral of the form

$$\int f(x')p(x'|\boldsymbol{y})\,dx'. \tag{3}$$

Applying Bayes' theorem, the integral can be written as

$$\int f(x')p(x'|\boldsymbol{y})\,dx' = \int f(x')\frac{p(\boldsymbol{y}|x')p(x')}{\int p(\boldsymbol{y}|x'')\,dx''}\,dx'.$$

The last integral can be approximated by a sum over an observation database $\{(\boldsymbol{y}_i, x_i)\}_{i=1}^n$ that is distributed according to the a priori distribution $p(x)$

$$\int f(x')p(x'|\boldsymbol{y})\,dx' \approx \frac{1}{C}\sum_{i=1}^n w_i(\boldsymbol{y})f(x_i).$$

with the normalization factor $C$ given by $C = \sum_{i=1}^n w_i(\boldsymbol{y})$. The weights $w_i(\boldsymbol{y})$ are given by the probability $p(\boldsymbol{y}|\boldsymbol{y}_i)$ of the observed measurement $\boldsymbol{y}$ conditional on the database measurement $\boldsymbol{y}_i$, which is usually assumed to be multivariate Gaussian with covariance matrix $\boldsymbol{S}_o$:

$$w_i(\boldsymbol{y}) \propto \exp\left\{-\frac{(\boldsymbol{y}-\boldsymbol{y}_i)^T\boldsymbol{S}_o^{-1}(\boldsymbol{y}-\boldsymbol{y}_i)}{2}\right\}.$$

By approximating integrals of the form (3), it is possible to estimate the expectation value and variance of the a posteriori
distribution by choosing $f(x) = x$ and $f(x) = (x - \mathcal{E}(x|\boldsymbol{y}))^2$, respectively. While this is suitable to represent Gaussian distributions, a more general representation of the a posteriori distribution can be obtained by estimating the corresponding CDF (c.f. Eq. (2)) using

$$F_{x|\boldsymbol{y}}(x) \approx \frac{1}{C}\sum_{x_i < x} w_i(\boldsymbol{y}). \tag{4}$$

### 2.3 Machine Learning

Neglecting uncertainties, the retrieval of a quantity $x$ from a measurement vector $\boldsymbol{y}$ may be viewed as a simple multiple regression task. In machine learning, regression problems are typically approached by training a parametrized model $f : \boldsymbol{x} \mapsto \boldsymbol{y}$ to predict a desired output $\boldsymbol{y}$ from given input $\boldsymbol{x}$. Unfortunately, the use of the variables $\boldsymbol{x}$ and $\boldsymbol{y}$ in machine learning is directly opposite to their use in inverse theory. For the remainder of this section the variables $\boldsymbol{x}$ and $\boldsymbol{y}$ will be used to denote, respectively, the input and output to the machine learning model to ensure consistency with the common notation in the field of machine
learning. The reader must keep in mind that the method is applied in the later sections to predict a retrieval quantity $x$ from a measurement $\boldsymbol{y}$.





### 2.3.1 Supervised learning and loss functions

Machine learning regression models are trained using supervised training, in which the model $f$ learns the regression mapping from a training set $\{\boldsymbol{x}_i, \boldsymbol{y}_i\}_{i=1}^n$ with input values $\boldsymbol{x}_i$ and expected output values $\boldsymbol{y}_i$. The training is performed by finding model parameters that minimize the mean of a given loss function $\mathcal{L}(f(\boldsymbol{x}), \boldsymbol{y})$ on the training set. The most common loss function for

regression tasks is the squared error loss

$$\mathcal{L}_{se}(f(\boldsymbol{x}), \boldsymbol{y}) = (f(\boldsymbol{x}) - \boldsymbol{y})^T (f(\boldsymbol{x}) - \boldsymbol{y}), \tag{5}$$

which trains the model $f$ to minimize the mean squared distance of the neural network prediction $f(\boldsymbol{x})$ from the expected output $\boldsymbol{y}$ on the training set. If the estimand $\boldsymbol{y}$ is a random vector drawn from a conditional probability distribution $p(\boldsymbol{y}|\boldsymbol{x})$, a regressor trained using a squared error loss function learns to predict the conditional expectation value of the distribution

$p(\boldsymbol{y}|\boldsymbol{x})$ (Bishop, 1994). Depending on the choice of the loss function, the regressor can also learn to predict other statistics of the distribution $p(\boldsymbol{y}|\boldsymbol{x})$ from the training data.

### 2.3.2 Quantile regression

Given the cumulative distribution function $F(x)$ of a probability distribution $p$, its $\tau$th quantile $x_\tau$ is defined as

$$x_\tau = \inf\{x \, : \, F(x) \geq \tau\}, \tag{6}$$

i.e. the greatest lower bound of all values of $x$ for which $F(x) \geq \tau$. As shown by Koenker (2005), the $\tau$th quantile $x_\tau$ of $F$ minimizes the expectation value $\mathcal{E}_x(\mathcal{L}_\tau(x_\tau, x)) = \int_{-\infty}^{\infty} \mathcal{L}_\tau(x_\tau, x') p(x') \, dx'$ of the function

$$\mathcal{L}_\tau(x_\tau, x) = \begin{cases} \tau|x - x_\tau|, & x_\tau < x \\ (1-\tau)|x - x_\tau|, & \text{otherwise.} \end{cases} \tag{7}$$

By training a machine learning regressor $f$ to minimize the mean of the quantile loss function $\mathcal{L}_\tau(f(\boldsymbol{x}), y)$ over a training set $\{\boldsymbol{x}_i, y_i\}_{i=1}^n$, the regressor learns to predict the quantiles of the conditional distribution $p(y|\boldsymbol{x})$. This can be extended to obtain

an approximation of the cumulative distribution function of $F_{y|\boldsymbol{x}}(y)$ by training the network to estimate multiple quantiles of $p(y|\boldsymbol{x})$.

### 2.3.3 Neural networks

A neural network computes a vector of output activations $\boldsymbol{y}$ from a vector of input activations $\boldsymbol{x}$. Feed-forward artificial neural networks (ANNs) compute the vector $\boldsymbol{y}$ by application of a given number of subsequent, learnable transformations to the input

activations $\boldsymbol{x}$:

$$\boldsymbol{x}_0 = \boldsymbol{x}$$

$$\boldsymbol{x}_i = f_i(\boldsymbol{W}_i \boldsymbol{x}_{i-1} + \boldsymbol{\theta}_i)$$

$$\boldsymbol{y} = \boldsymbol{x}_n.$$





The activation functions $f_i$ as well as the number and sizes of the hidden layers $\boldsymbol{x}_1, \ldots, \boldsymbol{x}_{n-1}$ are prescribed, structural parameters of a neural network model, generally referred to as hyperparameters. The learnable parameters of the model are the weight matrices $\boldsymbol{W}_i$ and bias vectors $\boldsymbol{\theta}_i$ of each layer. Neural networks can be efficiently trained in a supervised manner by using gradient based minimization methods to find suitable weights $\boldsymbol{W}_i$ and bias vectors $\boldsymbol{\theta}_i$. By using the mean of the quantile

loss function $\mathcal{L}_\tau$ as the training criterion, a neural network can be trained to predict the quantiles of the distribution $p(\boldsymbol{y}|\boldsymbol{x})$, thus turning the network into a quantile regression neural network.

### 2.3.4 Adversarial training

Adversarial training is a data augmentation technique that has been proposed to increase the robustness of neural networks to perturbations in the input data (Goodfellow et al., 2014). It has been shown to be effective also as a method to improve the

calibration of probabilistic predictions from neural networks (Lakshminarayanan et al., 2016). The basic principle of adversarial training is to augment the training data with perturbed samples that are likely to yield a large change in the network prediction. The method used here to implement adversarial training is the fast gradient sign method proposed by Goodfellow et al. (2014). For a training sample $(\boldsymbol{x}_i, \boldsymbol{y}_i)$ consisting of input $\boldsymbol{x}_i \in \mathrm{R}^n$ and expected output $\boldsymbol{y}_i \in \mathrm{R}^m$, the corresponding adversarial sample $(\tilde{\boldsymbol{x}}_i, \boldsymbol{y}_i)$ is chosen to be

$$\tilde{\boldsymbol{x}}_i = \boldsymbol{x}_i + \delta_{\mathrm{adv}} \mathrm{sign}\left(\frac{d\mathcal{L}(\hat{\boldsymbol{x}}(\boldsymbol{x}_i), x)}{d\boldsymbol{x}_i}\right), \tag{8}$$

i.e. the direction of the perturbation is chosen in such a way that it maximizes the absolute change in the loss function $\mathcal{L}$ due to an infinitesimal change in the input parameters. The adversarial perturbation factor $\delta_{\mathrm{adv}}$ determines the strength of the perturbation and becomes an additional hyperparameter of the neural network model.

### 2.4 Evaluating Probabilistic Predictions

A problem that remains is how to compare two estimates $p'(x|\boldsymbol{y}), p''(x|\boldsymbol{y})$ of a given a posteriori distribution against a single observed sample $x$ from the true distribution $p(x|\boldsymbol{y})$. A good probabilistic prediction for the value $x$ should be sharp, i.e. concentrated in the vicinity of $x$, but at the same time well calibrated, i.e. predicting probabilities that truthfully reflect observed frequencies (Gneiting et al., 2005). Summary measures for the evaluation of predicted conditional distributions are called scoring rules (Gneiting and Raftery, 2007). An important property of scoring rules is propriety, which formalizes the concept

of the scoring rule rewarding both sharpness and calibration of the prediction. Besides providing reliable measures for the comparison of probabilistic predictions, proper scoring rules can be used as loss functions in supervised learning to incentivize statistically consistent predictions.

The quantile loss function given in equation (7) is a proper scoring rule for quantile estimation and can thus be used to compare the skill of different methods for quantile estimation (Gneiting and Raftery, 2007). Another proper scoring rule

for the evaluation of an estimated cumulative distribution function $F$ against an observed value $x$ is the continuous ranked





probability score (CRPS):

$$\mathrm{CRPS}(F, x) = \int\limits_{-\infty}^{\infty} \left( F(x') - I_{x \leq x'} \right)^2 \, dx'. \tag{9}$$

Here, $I_{x \leq x'}$ is the indicator function that is equal to 1 when the condition $x \leq x'$ is true and 0 otherwise. For the methods used in this article the integral can only be evaluated approximately. The exact way in which this is done for each method is described in detail in Sect. 3.1.3, 3.1.4.

The scoring rules presented above evaluate probabilistic predictions against a single observed value. However, since MCMC simulations can be used to approximate the true a posteriori distribution to an arbitrary degree of accuracy, the probabilistic predictions obtained from BMCI and QRNN can be compared directly to the a posteriori distributions obtained using MCMC. In the idealized case where the modeling assumptions underlying the MCMC simulations are true, the sampling distribution obtained from MCMC will converge to the true posterior and can be used as a ground truth to assess the predictions obtained from the other methods.

### 2.4.1 Calibration plots

Calibration plots are a graphical method to assess the calibration of prediction intervals derived from probabilistic predictions. For a set of prediction intervals with probabilities $p = p_1, \ldots, p_n$, the fraction of cases for which the true value did lie within the bounds of the interval is plotted against the value $p$. If the predictions are well calibrated, the probabilities $p$ match the observed frequencies and the calibration curve is close to the diagonal $y = x$. An example of a calibration plot for three different predictors is given in Figure 1. Compared to the scoring rules described above, the advantage of the calibration curves is that they indicate whether the predicted intervals are too narrow or too wide. Predictions that overestimate the uncertainty yield intervals that are too wide and result in a calibration curve that lies above the diagonal, whereas observations underestimating the uncertainty will yield a calibration curve that lies below the diagonal.

## 3 Application to a synthetic retrieval case

In this section, a simulated retrieval of column water vapor from passive microwave observations is used to benchmark the performance of BMCI and QRNN against MCMC simulation. The retrieval case has been set up to provide an idealized but realistic scenario in which the true a posteriori distribution can be approximated using MCMC simulation. The MCMC results can therefore be used as the reference to investigate the retrieval performance of QRNNs and BMCI. Furthermore it is investigated how different hyperparameters influence the performance of the QRNN, and lastly how the size of the training set and retrieval database impact the performance of QRNNs and BMCI.



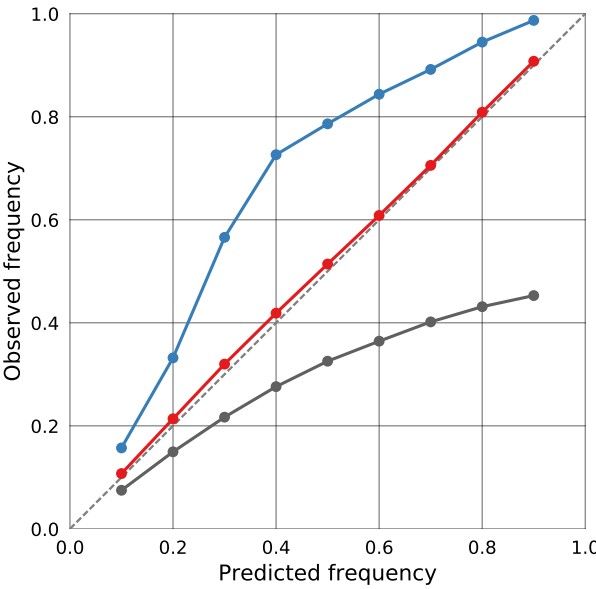

**Figure 1.** Example of a calibration plot displaying calibration curves for overly confident predictions (dark gray), well calibrated predictions (red), and overly cautious predictions (blue).

## 3.1 The retrieval

For this experiment, the retrieval of column water vapor (CWV) from passive microwave observations over the ocean is considered. The state of the atmosphere is represented by profiles of temperature and water vapor concentrations on 15 pressure levels between $10^3$ and $10$hPa. The variability of these quantities has been estimated based on ECMWF ERA Interim data (Dee
5   et al., 2011) from the year 2016, restricted to latitudes between $23°$ and $66°$ N. Parametrizations of the multivariate distributions of temperature and water vapor were obtained by fitting a joint multivariate normal distribution to the temperature and the logarithm of water vapor concentrations. The fitted distribution represents the a priori knowledge on which the simulations are based.

### 3.1.1 Forward model simulations

10  The Atmospheric Radiative Transfer Simulator (ARTS, Eriksson et al. (2011)) is used to simulate satellite observations of the atmospheric states sampled from the a priori distribution. The observations consist of simulated brightness temperatures from five channels around $23, 88, 165, 183$GHz (c.f. Table 1) of the ATMS sensor.

The simulations take into account only absorption and emission from water vapor. Ocean surface emissivities are computed using the FASTEM-6 model (Kazumori and English, 2015) with an assumed surface wind speed of zero. The sea surface




| Channel | Center frequency | Offset | Bandwidth |
|---|---|---|---|
| 1 | 23.8GHz | — | 270MHz |
| 2 | 88.2GHz | — | 500MHz |
| 3 | 165.5GHz | — | 300MHz |
| 4 | 183.3GHz | 7GHz | 2000MHz |
| 5 | 183.3GHz | 3GHz | 1000MHz |

**Table 1.** Observation channels used for the synthetic retrieval of column water vapor.

temperature is assumed to be equal to the temperature at the pressure level closest to the surface but no lower than 270K. Sensor characteristics and absorption lines are taken from the ATMS sensor descriptions that are provided within the ARTS XML Data package. Simulations are performed for a nadir looking sensor and neglecting polarization. The observation uncertainty is assumed to be independent Gaussian noise with a standard deviation of 1 K.

### 3.1.2 MCMC implementation

The MCMC retrieval is based on a Python implementation of the Metropolis algorithm (Gelman et al., 2013, Ch. 12) that has been developed within the context of this study. It is released as part of the *typhon: tools for atmospheric research* software package (The typhon authors, 2018).

The MCMC retrieval is performed in the space of atmospheric states described by the profiles of temperature and the logarithm of water vapor concentrations. The multivariate Gaussian distribution that has been obtained by fit to the ERA Interim data is taken as the a priori distribution. A random walk is used as the proposal distribution, with its covariance matrix taken as the a priori covariance matrix. A single MCMC retrieval consists of 8 independent runs, initialized with different random states sampled from the a priori distribution. Each run starts with a warm-up phase followed by an adaptive phase during which the covariance matrix of the proposal distribution is scaled adaptively to keep the acceptance rate of proposed states close to the optimal 21% (Gelman et al., 2013). This is followed by a production phase during which 5000 samples of the a posteriori distribution are generated. Only 1 out of 20 generated samples is kept in order to decrease the correlation between the resulting states. Convergence of each simulation is checked by computing the scale reduction factor $\hat{R}$ and the effective number of independent samples (Gelman et al., 2013, Eq. (11.12), (11.13)). The retrieval is discarded if the values are not smaller than 1.1 and larger than 100, respectively. Each MCMC retrieval generates a sequence of atmospheric states from which the column water vapor is obtained by integration of the water vapor concentration profile. The distribution of observed CWV values is then taken as the retrieved a posteriori distribution.

### 3.1.3 QRNN implementation

The implementation of quantile regression neural networks is based on the Keras Python package for deep learning (Chollet et al., 2015). It is also released part of the typhon package.





For the training of quantile regression neural networks, the quantile loss function $\mathcal{L}_\tau(x_\tau, x)$ has been implemented so that it can be used as a training loss function within the Keras framework. The function can be initialized with a sequence of quantile fractions $\tau_1, \ldots, \tau_k$ allowing the neural network to learn to predict the corresponding quantiles $x_{\tau_1}, \ldots, x_{\tau_k}$.

Custom data generators have been added to the implementation to incorporate information on measurement uncertainty into

the training process. If the training data is noise free, the data generator can be used to add noise to each training batch according to the assumptions on measurement uncertainty. The noise is added immediately before the data is passed to the neural network, keeping the original training data noise free. This ensures that the network does not see the same, noisy training sample twice during training, thus counteracting overfitting.

An adaptive form of stochastic batch gradient descent is used for the neural network training. During the training, loss is

monitored on a validation set. When the loss on the validation set hasn't decreased for a certain number of epochs, the training rate is reduced by a given reduction factor. The training stops when a predefined minimum learning rate is reached.

The reconstruction of the CDF from the estimated quantiles is obtained by using the quantiles as nodes of a piece-wise linear approximation and extending the first and last segments out to 0 and 1, respectively. This approximation is also used to compute the CRPS score on the test data.

### 3.1.4 BMCI implementation

The BMCI method has likewise been implemented in Python and added to the typhon package. In addition to retrieving the first two moments of the posterior distribution, the implementation provides functionality to retrieve the posterior CDF using Eq. (4). Approximate posterior quantiles are computed by interpolating the inverse CDF at the desired quantile values. To compute the CRPS score for a given retrieval, the trapezoidal rule is used to perform the integral over the values $x_i$ in the

retrieval database $\{\boldsymbol{y}_i, x_i\}_{i=1}^n$.

### 3.2 QRNN model selection

Just as with common neural networks, QRNNs have several hyperparameters that cannot be learned directly from the data, but need to be tuned independently. For this study the dependence of the QRNN performance on its hyperparameters has been investigated. The results are included here as they may be a helpful reference for future applications of QRNNs.

For this analysis, hyperparameters describing the structure of the QRNN model are investigated separately from training parameters. The hyperparameters describing the structure of the QRNN are:

1. the number of hidden layers,

2. the number of neurons per layer,

3. the type of activation function.

The training method described in Sect. 3.1.3 is defined by the following training parameters:

4. the batch size used for stochastic batch gradient descent,





5. the minimum learning rate at which the training is stopped,

6. the learning rate decay factor,

7. the number of training epochs without progress on the validation set before the learning rate is reduced.

### 3.2.1 Structural parameters

To investigate the influence of hyperparameters 1 - 3 on the performance of the QRNN, 10-fold cross validation on the training set consisting of $10^6$ samples has been used to estimate the performance of different hyperparameter configurations. As performance metric the mean quantile loss on the validation set averaged over all predicted quantiles for $\tau = 0.05, 0.1, 0.2, \ldots, 0.9, 0.95$ is used. A grid search over a subspace of the configuration space was performed to find optimal parameters. The results of the analysis are displayed in Figure 2. For the configurations considered, the layer width has the

most significant effect on the performance. Nevertheless, only small performance gains are obtained by increasing the layer width to values above 64 neurons. Another general observation is that networks with three hidden layers generally outperform networks with fewer hidden layers. Networks using ReLU activation functions not only achieve slightly better performance than networks using tanh or sigmoid activation functions, but also show significantly lower variability. Based on these results, a neural network with three hidden layers, 128 neurons in each layer and ReLU activation functions has been selected for the

comparison to BMCI.

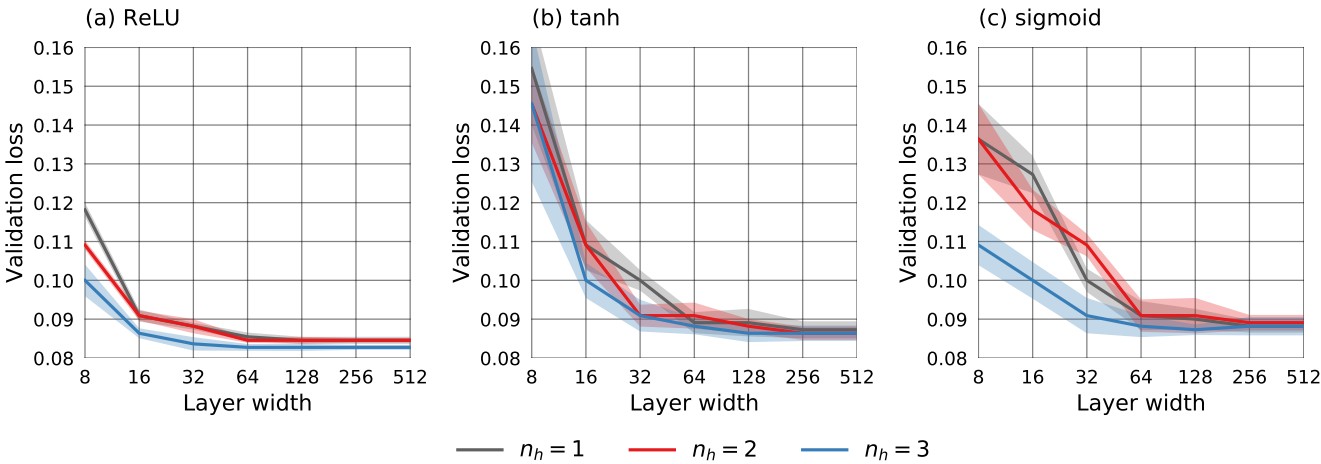

**Figure 2.** Mean validation set loss (solid lines) and standard deviation (shading) of different hyperparameter configurations with respect to layer width (number of neurons). Different lines display the results for different numbers of hidden layers $n_h$. The three panels show the results for ReLU, tanh, and sigmoid activation functions.





### 3.2.2 Training parameters

For the optimization of the training parameters 4 - 7, a very coarse grid search was performed, using only three different values for each parameter. In general, the training parameters showed only little effect ($< 2\%$ for the combinations considered here) on the QRNN performance compared to the structural parameters. The best cross-correlation performance was obtained for

slow training with a small learning rate reduction factor of $1.5$ and decreasing the learning rate only after $10$ training epochs without reduction of the validation loss. No significant increase in performance could be observed for values of the learning rate minimum below $10^{-4}$. With respect to the batch size, the best results were obtained for a batch size of $128$ samples.

### 3.3 Comparison against MCMC

In this section, the performance of a single QRNN and an ensemble of 10 QRNNs are analyzed. The predictions from the en-

semble are obtained by averaging the predictions from each network in the ensemble. All tests in this subsection are performed for a single QRNN, the ensemble of QRNNs, and BMCI. The retrieval database used for BMCI and the training of the QRNNs in this expirement consists of $10^6$ entries.

Figure 3 displays retrieval results for eight example cases. The choice of the cases is based on the Kolmogorov-Smirnov (KS) statistic, which corresponds to the maximum absolute deviation of the predicted CDF from the reference CDF obtained

by MCMC simulation. A small KS value indicates a good prediction of the true CDF, while a high value is obtained for large deviations between predicted and reference CDF. The cases shown correspond to the 10th, 50th, 90th and 99th percentile of the distribution of KS values obtained using BMCI or a single QRNN. In this way they provide a qualitative overview of the performance of the methods.

In the displayed cases, both methods are generally successful in predicting the a posteriori distribution. Only for the 99th

percentile of the KS value distribution does the BMCI prediction show significant deviations from the reference distribution. The jumps in the estimated a posteriori CDF indicate that the deviations are due to undersampling of the input space in the retrieval database. This results in overproportionally high weights attributed to the few entries close to the observation. For this specific case the QRNN provides a better estimate of the a posteriori CDF even though both predictions are based on the same data.

To obtain a more comprehensive view on the performance of QRNNs and BMCI, the predictions obtained from both methods are compared to those obtained from MCMC for 6500 test cases. For the comparison, let the *effective quantile fraction* $\tau_{\text{eff}}$ be defined as the fraction of MCMC samples that are less than or equal to the predicted quantile $\widehat{x_\tau}$ obtained from QRNN or BMCI. In general, the predicted quantile $\widehat{x_\tau}$ will not correspond exactly to the true quantile $x_\tau$, but rather an effective quantile $x_{\tau_{\text{eff}}}$, defined by the fraction $\tau_{\text{eff}}$ of the samples of the distribution that are smaller than or equal to the predicted value $\widehat{x_\tau}$. The

resulting distributions of the effective quantile fractions for BMCI and QRNNs are displayed in Figure 4 for the estimated quantiles for $\tau = 0.1, 0.2, \ldots, 0.9$.

For an ideal estimator of the quantile $x_\tau$, the resulting distribution would be a delta function centered at $\tau$. Due to the estimation error, however, the $\tau_{\text{eff}}$ values are distributed around the true quantile fraction $\tau$. The results show that both BMCI





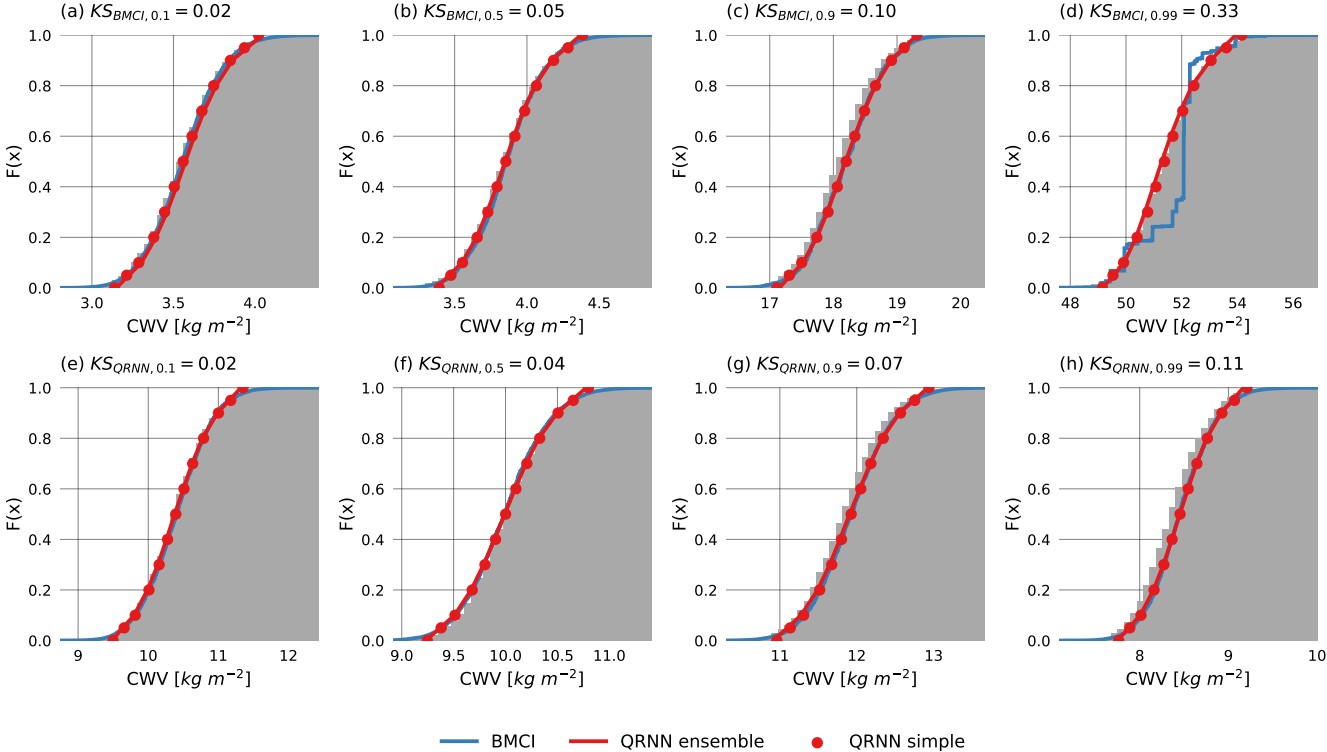

**Figure 3.** Retrieved a posteriori CDFs obtained using MCMC (grey), BMCI (blue), a single QRNN (red line) and an ensemble of QRNNs (red marker). Cases displayed in the first row correspond to the 1st, 50th, 90th, and 99th percentiles of the distribution of the Kolmogorov-Smirnov statistic of BMCI compared to the MCMC reference. The second row displays the same percentiles of the distribution of the Kolmogorov-Smirnov statistic of the single QRNN predictions compared to MCMC.

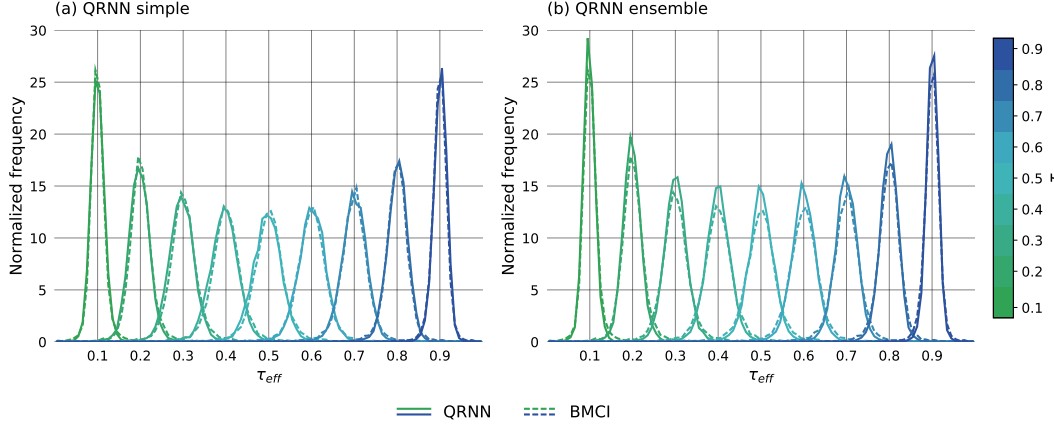

**Figure 4.** Distribution of effective quantile fractions $\tau_{\text{eff}}$ achieved by QRNN and BMCI on the test data. The left plot displays the performance of a single QRNN compared to BMCI, the right plot the performance of the ensemble.





and QRNN provide fairly accurate estimates of the quantiles of the a posterior distribution. Furthermore, all methods yield equally good predictions, making the distributions virtually identical.

## 3.4 Training set size impact

Finally, we investigate how the size of the training data set used in the training of the QRNN (or as retrieval database for BMCI)

affects the performance of the retrieval method. This has been done by randomly generating training subsets from the original training data with sizes logarithmically spaced between $10^3$ and $10^6$ samples. For each size, five random training subsets have been generated and used to retrieve the test data with a single QRNN and BMCI. As test data, a separate test set consisting of $10^5$ simulated observations vectors and corresponding CWV values is used.

Figure 5 displays the means of the mean absolute percentage error (MAPE, Panel (a)) and the mean continuous ranked

probability score (CRPS, Panel (b)) achieved by both methods on the differently sized training sets. For the computation of the MAPE, the CWV prediction is taken as the median of the estimated a posteriori distribution obtained using QRNNs or BMCI. This value is compared to the true CWV value corresponding to the atmospheric state that has been used in the simulation. As expected, the performance of both methods improves with the size of the training set. With respect to the MAPE, both methods perform equally well for a training set size of $10^6$, but the QRNN outperforms BMCI for all smaller training set sizes. With

respect to CRPS, a similar behavior is observed. These are reassuring results, as they indicate that not only the accuracy of the predictions (measured by the MAPE and CRPS) improves as the amount of training data increases, but also their calibration (measured only by the CRPS).

Finally, the mean of the quantile loss $\mathcal{L}_\tau$ on the test set for $\tau = 0.1, 0.5, 0.9$ has been considered (Figure 6). Qualitatively, the results are similar to the ones obtained using MAPE and CRPS. The QRNN outperforms BMCI for smaller training set sizes

but converges to similar values for training set sizes of $10^6$.

The results presented in this section indicate that QRNNs can, at least under idealized conditions, be used to estimate the a posteriori distribution of Bayesian retrieval problems. Moreover, they were shown to work equally well as BMCI for large data sets. What is interesting is that for smaller data sets, QRNNs even provide better estimates of the a posteriori distribution than BMCI. This indicates that QRNNs provide a better representation of the functional dependency of the a posteriori distribution

on the observation data, thus achieving better interpolation in the case of scarce training data. Nonetheless, it remains to be investigated, if this advantage can also be observed for real world data.

## 4    Retrieving cloud top pressure from MODIS using QRNNs

In this section QRNNs are applied to retrieve cloud top pressure (CTP) using observations from the moderate resolution imaging spectroradiometer (MODIS Platnick et al. (2003)). This experiment is based on the work by Håkansson et al. (2018)

who developed a cloud top pressure retrieval algorithm (NN-CTTH) based on neural networks. A QRNN based CTP retrieval is compared to the NN-CTTH algorithm and it is investigated how QRNNs can be used to estimate the retrieval uncertainty.





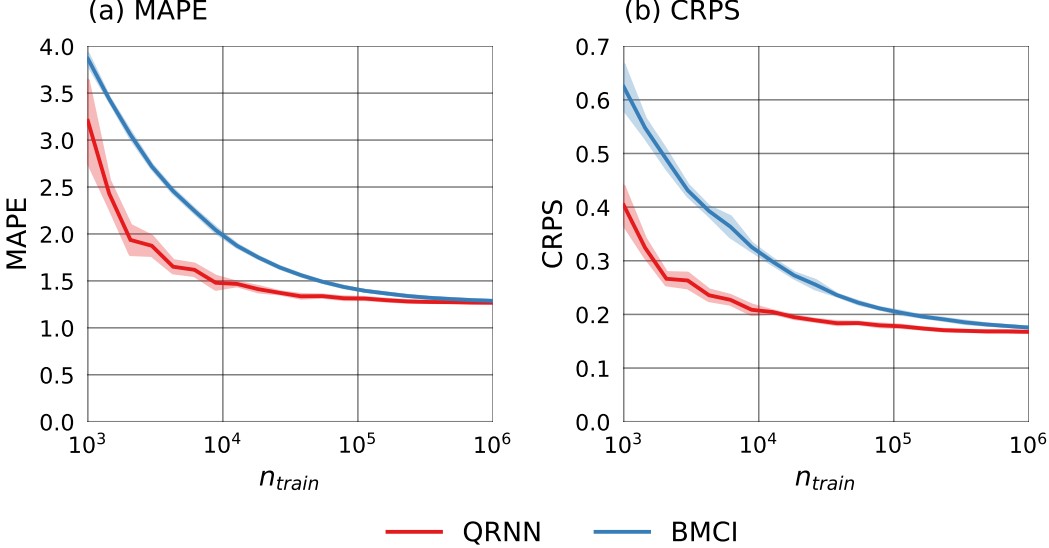

**Figure 5.** MAPE (Panel (a)) and CRPS (Panel (b)) achieved by QRNN (red) and BMCI (blue) on the test set using differently sized training sets and retrieval databases. For each size, five random subsets of the original training data were generated. The lines display the means of the observed values. The shading indicates the range of $\pm\sigma$ around the mean.

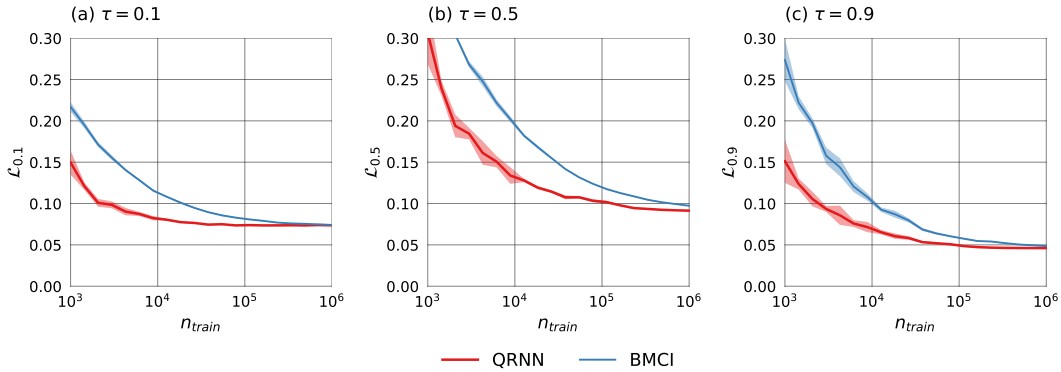

**Figure 6.** Mean quantile loss for different training set sizes $n_{\text{train}}$ and $\tau = 0.1, 0.5, 0.9$.





## 4.1 Data

Exactly the same data as for the training of the NN-CTTH algorithm is used for the training of the QRNNs. The data set
consists of MODIS Level 1B data (MODIS Characterization Support Team / MODIS Adaptive Processing System, 2015a,
b) collocated with cloud properties obtained from CALIOP (Winker et al., 2009). The *top layer pressure* variable from the
CALIOP data is used as retrieval target. The data was taken from all orbits from 24 days (the 1st and 14th of every month)
from the year 2010. Håkansson et al. (2018) train neural networks using different combinations of input features and compare
the resulting performance. Only the 11μm and 12μm channels of MODIS are considered here, as they were found to yield a
good compromise of performance and flexibility. In addition to the single pixel input, structural information is provided to the
neural network in the form of various statistics computed on a $5 \times 5$ neighborhood around the pixel. Furthermore, numerical
weather prediction data is provided to the network in the form of surface pressure, temperatures at several pressure levels, and
column integrated water vapor.

The data used for the training of the QRNNs are the *training* and *during-training validation set* from Håkansson et al. (2018).
The QRNNs are compared to the AVHRR version of the NN-CTTH algorithm, as this is the one using the same input data for
the retrieval. The comparison to NN-CTTH is performed on the data set for *testing under development*.

## 4.2 Training

The same training scheme as described in Sect. 3.1.3 is used for the training of the QRNNs. The *during-training validation*
set is used to monitor training progress based on which the learning rate is reduced or the training stopped. After performing a
grid search (results not shown) over width, depth and minibatch size, the best performance on the validation set was obtained
for networks with four layers with 64 neurons each, ReLU activation functions, and a batch size of 128 samples.

The main difference in the training process compared to the previous experiment is how measurement uncertainties are
incorporated. For the simulated retrieval, the training data was noise-free, so measurement uncertainties could be realistically
represented by adding noise according to the sensor characteristics. This is not the case for MODIS observations; instead,
adversarial training is used here to ensure well-calibrated predictions. For the tuning of the perturbation parameter $\delta_{\mathrm{adv}}$ (c.f.
Sect. 2.3.4), the calibration on the during-training validation set was monitored using a calibration plot. Ideally, it would be
desirable to use a separate data set to tune this parameter, but this was sufficient in this case to achieve good results on the
test data. The calibration curves obtained using different values of $\delta_{\mathrm{adv}}$ are displayed in Figure 7. It can be seen from the plot
that without adversarial training ($\delta_{\mathrm{adv}} = 0$) the predictions obtained from the QRNN are overly confident, leading to prediction
intervals that underrepresent the uncertainty in the retrieval. Since adversarial training may be viewed as a way of representing
observation uncertainty in the training data, larger values of $\delta_{\mathrm{adv}}$ lead to less confident predictions. Based on these results,
$\delta_{\mathrm{adv}} = 0.05$ is chosen for the training.

Except for the use of adversarial training, the structure of the underlying network and the training process of the QRNN are
fairly similar to what is used for the NN-CTTH retrieval. The QRNN uses four instead of two hidden layers with 64 neurons
in each of them instead of 30 in the first and 15 in second layer. While this makes the neural network used in the QRNN





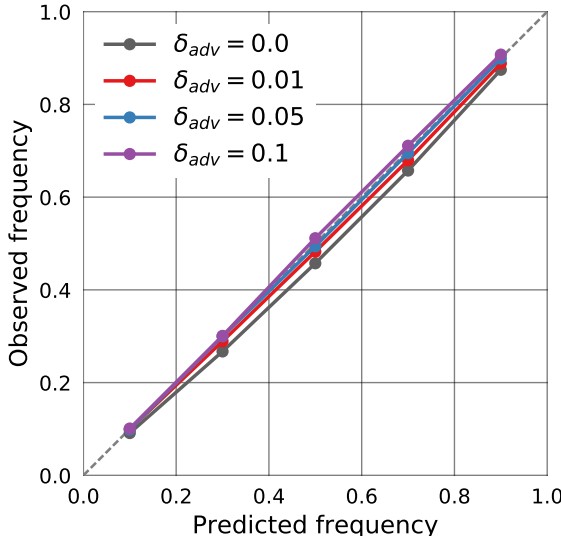

**Figure 7.** Calibration of the QRNN prediction intervals on the validation set used during training. The curves display the results for no adversarial training ($\delta_{\mathrm{adv}} = 0$) and adversarial training with perturbation factor $\delta_{\mathrm{adv}} = 0.01, 0.05, 0.1$.

slightly more complex, this should not be a major drawback since computational performance is generally not critical for neural network retrievals.

### 4.3 Prediction accuracy

Figure 8 displays the error distributions of the predicted CTP values on the *testing during development* data set. The error
is plotted separately for low, medium and high clouds (as classified by the CALIOP feature classification flag) as well as the complete data set. For the QRNNs, the prediction is taken as the median of the a posteriori distribution. Both the simple QRNN and the ensemble of QRNNs perform slightly better than the NN-CTTH algorithm for low and high clouds. For medium clouds, no significant difference in the performance of the methods can be observed. The ensemble of QRNNs seems to slightly improve upon the prediction accuracy of a single QRNN but the difference is likely negligible.

### 4.4 Uncertainty estimation

The NN-CTTH algorithm retrieves CTP but does not provide case-specific uncertainty estimates. Instead, an estimate of uncertainty is provided in the form of the observed mean absolute error on the test set. In order to compare these uncertainty estimates with those obtained using QRNNs, Gaussian error distributions are fitted to the observed error based on the observed mean absolute error (MAE) and mean squared error (MSE). A Gaussian error model is chosen here as it is arguably the most
common distribution used to represent random errors.





**Figure 8.** Error distributions of predicted CTP values (CTP$_{pred}$) with respect to CTP from CALIOP (CTP$_{ref}$) for different cloud types and the complete test set.




A plot of the errors observed on the *testing during development* data set and the fitted Gaussian error distributions is displayed in Panel (a) of Figure 9. The fitted error curves correspond to the Gaussian probability density functions with the same MAE and MSE as observed on the test data. Panel (b) displays the observed error together with the predicted error obtained from a single QRNN. The predicted error is computed as the deviation of a random sample of the estimated a posteriori distribution

from its median. The fitted Gaussian error distributions clearly do not provide a good fit to the observed error. On the other hand, the predicted errors obtained from the QRNN a posteriori distributions yield good agreement with the observed error. This indicates that the QRNN successfully learned to predict retrieval uncertainties. Furthermore, the results show that the ensemble of QRNNs actually provides a slightly worse fit to the observed error than a single QRNN. An ensemble of QRNNs thus does not necessarily improve the calibration of the predictions.

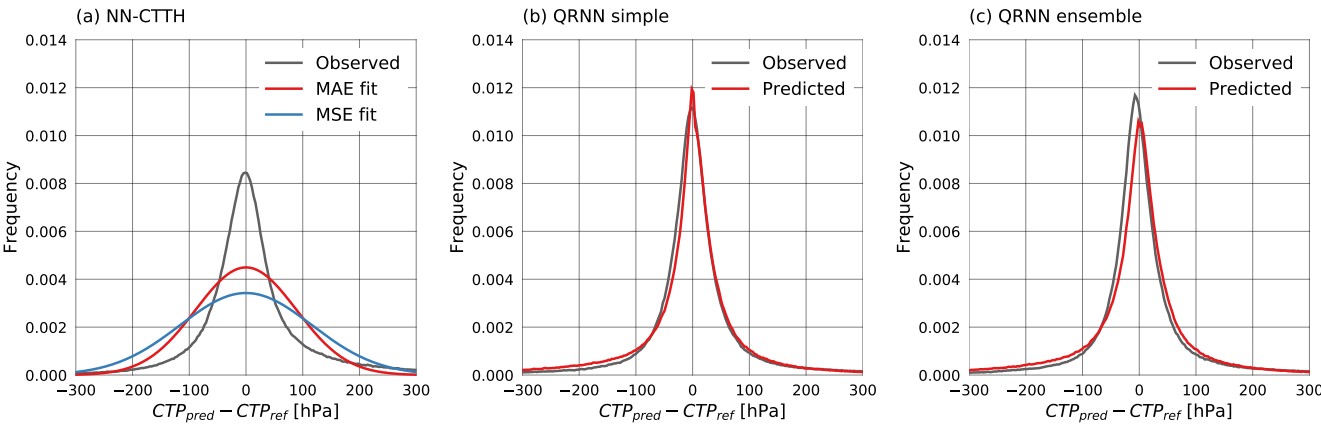

**Figure 9.** Predicted and observed error distributions. Panel (a) displays the observed error for the NN-CTTH retrieval as well as the Gaussian error distributions that have been fitted to the observed error distribution based on the MAE and MSE. Panel (b) displays the observed test set error for a single QRNN as well as the predicted error obtained as the deviation of a random sample of the predicted a posteriori distribution from the median. Panel (c) displays the same for the ensemble of QRNNs.

The Gaussian error model based on the MAE fit has also been used to produce prediction intervals for the CTP values obtained from the NN-CTTH algorithm. Figure 10 displays the resulting calibration curves for the NN-CTTH algorithm, a simple QRNN and an ensemble of QRNNs. The results support the finding that a single QRNN is able to provide well calibrated probabilistic predictions of the a posteriori distribution. The calibration curve for the ensemble predictions is virtually identical to that for the single network. The NN-CTTH predictions using a Gaussian fit are not as well calibrated and tend to provide

prediction intervals that are too wide for $p = 0.1, 0.3, 0.5, 0.7$ but overly narrow intervals for $p = 0.9$.

## 4.5 Sensitivity to a priori distribution

As shown above, the predictions obtained from the QRNN are statistically consistent in the sense that they predict probabilities that match observed frequencies when applied to test data. This however requires that the test data is statistically consistent





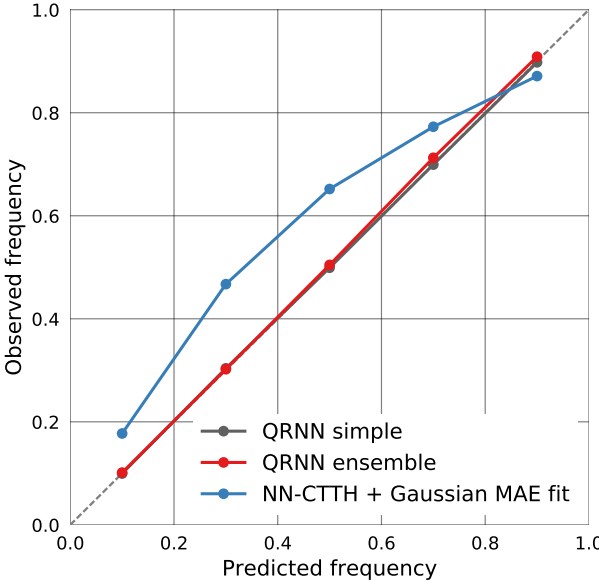

**Figure 10.** Calibration plot for prediction intervals derived from the Gaussian error model for the NN-CTTH algorithm (blue), the single QRNN (dark gray) and the ensemble of QRNNs (red).

with the training data. Statistically consistent here means that both data sets come from the same generating distribution, or in more Bayesian terms, the same a priori distribution. What happens when this is not the case can be seen when the calibration with respect to different cloud types is computed. Figure 11 displays calibration curves computed separately for low, medium and high clouds. As can be seen from the plot, the QRNN predictions are no longer equally well calibrated. Viewed from the Bayesian perspective, this is not very surprising as CTP values for median clouds have a significantly different a priori distribution compared to CTP values for all cloud types, thus giving different a posteriori distributions.

For the NN-CTTH algorithm, the results look different. While for low clouds the calibration deteriorates, the calibration is even slightly improved for high clouds. This is not surprising as the Gaussian fit may be more appropriate on different subsets of the test data.

## 5 Conclusions

In this article, quantile regression neural networks have been proposed as a method to estimate a posteriori distributions of Bayesian remote sensing retrievals. They have been applied to two retrievals of scalar atmospheric variables. It has been demonstrated that QRNNs are capable of providing accurate and well-calibrated probabilistic predictions in agreement with the Bayesian formulation of the retrieval problem.



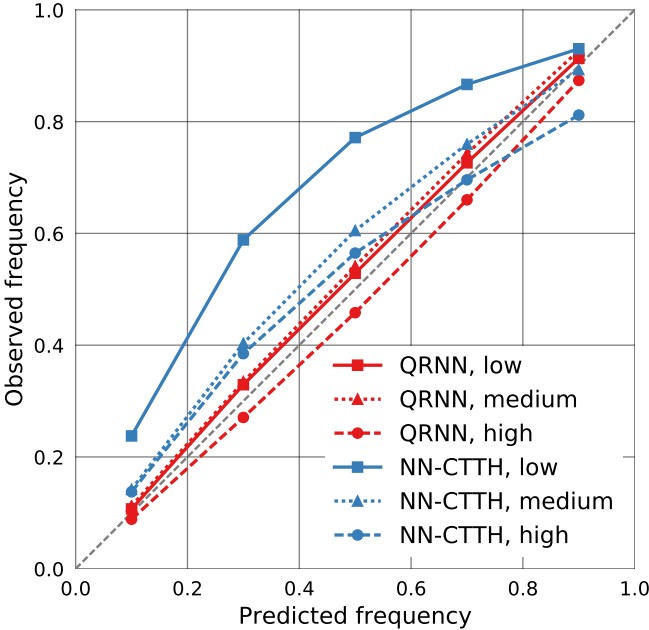

**Figure 11.** Calibration of the prediction intervals obtained from NN-CTTH (blue) and a single QRNN (red) with respect to specific cloud types.

The synthetic retrieval case presented in Sect. 3 shows that the conditional distribution learned by the QRNN is the same as the Bayesian a posteriori distribution obtained from methods that are directly based on the Bayesian formulation. This in itself seems worthwhile to note, as it reveals the importance of the training set statistics that implicitly represent the a priori knowledge. On the synthetic data set, QRNNs compare well to BMCI and even perform better for small data sets. This indicates that they are able to handle the "curse of dimensionality" (Friedman et al., 2001) better than BMCI, which would make them more suitable for the application to retrieval problems with high-dimensional measurement spaces.

While the optimization of computational performance of the BMCI method has not been investigated in this work, at least compared to a naive implementation of BMCI, QRNNs allow for at least one order of magnitude faster retrievals. QRNN retrievals can be easily parallelized and hardware optimized implementations are available for all modern computing architectures, thus providing very good performance out of the box.

Based on these very promising results, the next step in this line of research should be to compare QRNNs and BMCI on a real retrieval case to investigate if the findings from the simulations carry over to the real world. If this is the case, significant reductions in the computational cost of operational retrievals and maybe even better retrieval performance could be achieved using QRNNs.

In the second retrieval application presented in this article, QRNNs have been used to retrieve cloud top pressure from MODIS observations. The results show that not only are QRNNs able to improve upon state-of-the-art retrieval accuracy but




they can also learn to predict retrieval uncertainty. The ability of QRNNs to provide statistically consistent, case-specific uncertainty estimates should make them a very interesting alternative to non-probabilistic neural network retrievals. Nonetheless, also the sensitivity of the QRNN approach to a priori assumptions has been demonstrated. The posterior distribution learned by the QRNN depends on the validity of the a priori assumptions encoded in the training data. In particular, accurate uncertainty estimates can only be expected if the retrieved observations follow the same distribution as the training data. This, however, is a limitation inherent to all empirical methods.

The second application case presented here demonstrated the ability of QRNNs to represent non-Gaussian retrieval errors. While, as shown in this study, this is also the case for BMCI (Eq. (4)), it is common in practice to estimate only mean and standard deviation of the a posteriori distribution. Furthermore, implementations usually assume Gaussian measurement errors, which is an unlikely assumption if the observations in the retrieval database contain modeling errors. By requiring no assumptions whatsoever on the involved uncertainties, QRNNs may provide a more suitable way of representing (non-Gaussian) retrieval uncertainties.

The application of the Bayesian framework to neural network retrievals opens the door to a number of interesting applications that could be pursued in future research. It would for example be interesting to investigate if the a priori information can be separated from the information contained in the retrieved measurement. This would make it possible remove the dependency of the probabilistic predictions on the a priori assumptions, which can currently be considered a limitation of the approach. Furthermore, estimated a posteriori distributions obtained from QRNNs could be used to estimate the information content in a retrieval following the methods outlined by Rodgers (2000).

In this study only the retrieval of scalar quantities was considered. Another aspect of the application of QRNNs to remote sensing retrievals that remains to be investigated is how they can be used to retrieve vector-valued retrieval quantities. While the generalization to marginal, multivariate quantiles should be straight forward, it is unclear whether a better approximation of the quantile contours of the joint a posteriori distribution can be obtained using QRNNs.

*Code availability.* The implementation of the retrieval methods that were used in this article have been published as parts of the *typhon: tools for atmospheric research* (The typhon authors, 2018) software package. The source code for the calculations presented in Sect. 3 and 4 are accessible from public repositories (Pfreundschuh, 2018a, b).

*Author contributions.* All authors contributed to the study through discussion and feedback. Patrick Eriksson and Bengt Rydberg proposed the application of QRNNs to remote sensing retrievals. The study was designed and implemented by Simon Pfreundschuh, who also prepared the manuscript including figures, text and tables. Anke Thoss and Nina Håkansson provided the training data for the cloud top pressure retrieval.

*Competing interests.* The authors declare that they have no conflict of interest.



*Acknowledgements.* The scientists at Chalmers University of Technology were funded by the Swedish National Space Board.

The authors would like to acknowledge the work of Ronald Scheirer and Sara Hörnquist who were involved in the creation of the collocation dataset that was used as training and test data for the cloud top pressure retrieval.

Numerous free software packages were used to perform the numerical experiments presented in this article and visualize their results.

5    The authors would like to acknowledge the work of all the developers that contributed to making these tools freely available to the scientific community, in particular the work by Hunter (2007); Pérez and Granger (2007); Walt et al. (2011); The Python Language Foundation (2018).





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
