# Peer review of "A neural network approach to estimate a posteriori distributions of Bayesian retrieval problems"

_Atmospheric Measurement Techniques, 2018_

## Referee Comment (RC1) · C. Kummerow (Referee) · 7 May 2018

The paper provides an important development in neural network algorithms, and is extremely well written. I enjoyed reading it and it certainly deserves to be published. In fact, I have only one minor issue and some trivial editorial comments.

P. 15: in the comparisons between QRNN and the BMCI, as the training data or a-prior get smaller, the BMCI uncertainties need to be increased beyond the sensor noise to account for a sparse a-priori. If that was not done, it likely explains the divergence in the performance for smaller training sets. That said, finding the uncertainty due to a sparse a-priori is not at all trivial so it might still be an advantage for the QRNN but

perhaps slightly different than presented. A bit more explanation by the author on this topic would help the paper. The conclusion mentions this as well.

Editorial comments:

p.5, line 20: I really appreciate the author's explanation of why he used x and y in different ways for the two sections. I appreciate why he did that and the explanation makes clear what could otherwise have been a very confusing bit of text.

p. 12, line 12: Maybe I missed it but I don't think Rectilinear Linear Unit was ever defined in the paper.

I am quite certain that neither "Gaussianity" (p.3, line 2) nor "overproportionally" (p. 12, line 20) are real words.

p. 4, line 15: There is an extra "from" in front of "directly"

––––––––––––––––––––––––––––––

---

## Referee Comment (RC2) · Anonymous Referee #2 · 15 Jun 2018

Summary:

This paper introduces a new neural network retrieval approach that is capable of obtaining an a posteriori distribution of retrievals – providing an estimate of uncertainty in retrieved parameters. While the development of the algorithm is the primary focus of this paper, there is also an extensive effort to compare this approach to existing Bayesian approaches.

General Comments:

This paper represents a substantive contribution to the usage of machine learning techniques in remote sensing. The introduction of estimates of uncertainty in this im-

plementation increases the value of Neural Network solutions to remote sensing problems. In particular, the extensive comparison between Bayesian retrieval approaches and the QRNN highlights the power of this approach. It is particularly promising that this QRNN approach can provide insight into non-gaussian retrieval uncertainties, anything that challenges the reductive gaussian paradigm is fine in my book.

This paper certainly deserves to be published, but I have just a few minor concerns I think may help to make the paper more accessible to readers.

1) While I think that examining bias histograms of retrieval variables is a useful way of evaluating retrieval quality, it would be useful to provide readers a picture of the behavior of the original histograms. To that end, I think that, in addition to figure 8, a 1-D histogram (pressure on the vertical axis) of predicted cloud top pressures should be shown (including each of the 3 approaches).

2) I understand the usage of the CDF's in figure 3 to demonstrate the statistical value of the retrievals, but again, I think you need to directly show at least an example of a 1-D histogram of the retrieved variable. It gives readers a more direct sense of the variable being retrieved and provides context.

Specific Comments:

Irrespective of the outcome of this review process, I hope these specific comments can help the author improve their manuscript.

1. Page 2, Line 31: Abbreviations of MAP and 1DVAR are undefined.

2. Page 4, Line 15: The sentence at the beginning of section 2.2 would almost be better suited as a conclusory sentence at the end of section 2.1. As a matter of formatting, I think it's best to avoid a section being a single sentence long.

3. Page 9, Line 5: Can you provide an explanation for this restriction of latitudes in the dataset?

4. Page 10, Line 8: The formatting of this citation for Typhon seems like it might be incomplete or incorrect for the type of source being cited.

5. Page 10, Line 19: I don't follow the statement, "not smaller than and larger than 100, respectively". I read that as a logical statement that can't be possible because of the usage of "and." Do you perhaps mean "or?"

6. Page 10, Line 24: Missing word. Need to insert as into, "It is also released as part of the typhon package.

7. Page 15, Line 30: NN-CTTH is an undefined abbreviation. Because of the similarity to the abbreviation for cloud top height (CTH) I almost didn't notice that I didn't understand the name of the algorithm until later in the paper.

8. Page 17, Line 2: This feels like an incomplete sentence. Perhaps if you reorder it a bit it will make more sense. For example: "The NN-CTTH algorithm and the QRNNN are trained using the exact same data set. This training set consists. . ."

9. Page 17, Line 7: I also recommend reorganizing this sentence: "In Håkansson et al. (2018) neural networks are trained using different combinations of input features (e.g., [example from paper goes here]) in order to evaluate network performance with different inputs."

10. Page 17, Line 12-14: I don't follow along with this paragraph. Is the "testing under development" dataset the same as the AVHRR version? Could you clarify this paragraph some? Right now it seems that the confusion stems from the discussion of two different datasets within the space of only three sentences – making I difficult to clarify the difference between them. Is the first sentence referring to the first paragraph?

11. Page 17, Line 16-17: This sentence is a little confusing to read.

12. Page 18, Line 6-7: Could you expand on how the performance for low and high clouds differs here? Because you're discussing in terms of low and high clouds and cloud top pressure it's useful to highlight that a negative CTPpred-CTPref means that

the cloud is higher than the reference. Basically, for the NN-CTTH low clouds have a high-bias and high clouds have a low-bias. It's useful to say that explicitly.

13. Page 23, Line 20: For the sake of clarity for a more general audience can you provide an example of a vector-valued retrieval quantity.
* * *

---

## Author Comment (AC1) · 26 Jun 2018

**Referee comment:**

P. 15: in the comparisons between QRNN and the BMCI, as the training data or a-prior get smaller, the BMCI uncertainties need to be increased beyond the sensor noise to account for a sparse a-priori. If that was not done, it likely explains the divergence in the performance for smaller training sets. That said, finding the uncertainty due to a sparse a-priori is not at all trivial so it might still be an advantage for the QRNN but perhaps slightly different than presented. A bit more explanation by the author on this topic would help the paper. The conclusion mentions this as well.

**Author response:**

This is a very valid point that has indeed not been considered in the presented calculations. However, in particular since there is no formal way of doing this, it seems that finding suitable ways of handling scarce databases with BMCI would merit a study of its own. Applying just any ad-hoc solution to increase the measurement uncertainty is unlikely to do the BMCI method justice, so the authors judge it to be out of the scope of the study to investigate this further.
To address this in the manuscript, the following paragraph has been added:

 *A possible approach to handling scarce retrieval databases with BMCI is to artificially increase the assumed measurement uncertainty. This has not been performed for the BMCI results presented here and may improve the performance of the method. The difficulty with this approach is that the method formulation is based on the assumption of a sufficiently large database and thus can, at least formally, not be handle scarce training data. Finding a suitable way to increase the measurement uncertainty would thus require either additional methodological development or invention of an heuristic approach, both of which are outside the scope of this study.*

**Referee comment**

P. 12, line 12: Maybe I missed it but I don't think Rectilinear Linear Unit was ever defined in the paper.

**Author response**

The ReLU activation function is now introduced as Rectified Linear Unit the first time the acronym is used in the text.

**Referee comment**

I am quite certain that neither "Gaussianity" (p.3, line 2) nor "overproportionally" (p. 12, line 20) are real words.

**Author response**

The word *Gaussianity* has been replaced. The sentence now reads:

*Nonetheless, even neglecting the validity of the assumptions of Gaussian a priori and measurement errors as well as linearity of the forward model, the method is unsuitable for retrievals that involve complex radiative processes.*

Similarly, *overproportionally* (which correctly should have been overpropotionately) has been replaced by *excessively*.

**Referee comment**

p. 4, line 15: There is an extra "from" in front of "directly"

**Author comment**

The word *from* has been removed.

---

## Author Comment (AC2)

**1 General comments**

**Referee comment:**

5  1) While I think that examining bias histograms of retrieval variables is a useful way of evaluating retrieval quality, it would be useful to provide readers a picture of the behavior of the original histograms. To that end, I think that, in addition to figure 8, a 1-D histogram (pressure on the vertical axis) of predicted cloud top pressures should be shown (including each of the 3 approaches).

**Author response:**

10  The following figure has been added to the manuscript:

[Figure]

**Figure 1.** Distributions of predicted CTP values ($CTP_{pred}$) for high clouds in panel (a), medium clouds in panel (b), low clouds in panel (c) and the complete test set in panel (d).

Furthermore, the accompanying discussion of the single value retrieval results has been extended and now reads:

*Most data analysis will likely require a single predicted value for the cloud top pressure. To derive a point value from the QRNN prediction, the median of the estimated a posteriori distribution is used.*

*The distributions of the resulting median pressure values on the testing during development data set are displayed in Figure 8 together with the retrieved pressure values from the NN-CTTH algorithm. The distributions are displayed separately for low, medium and high clouds (as classified by the CALIOP feature classification flag) as well as the complete data set. From these results it can be seen that the values predicted by the QRNN have stronger peaks low in the atmosphere for low clouds and high in the atmosphere for high clouds. For medium clouds the peak is more spread out and has heavier tails low and high in the atmosphere than the values retrieved by the NN-CTTH algorithm.*

*Figure 9 displays the error distributions of the predicted CTP values on the testing during development data set, again separated by cloud type as well as the complete data set. Both the simple QRNN and the ensemble of QRNNs perform slightly better than the NN-CTTH algorithm for low and high clouds. For medium clouds, no significant difference in the performance of the methods can be observed. The ensemble of QRNNs seems to slightly improve upon the prediction accuracy of a single QRNN but the difference is likely negligible. Compared to the QRNN results the CTP predicted by NN-CTTH is biased low for low clouds and biased high for high clouds.*

*Even though both the QRNN and the NN-CTTH retrieval use the same input and training data, the predictions from both retrievals differ considerably. Using the Bayesian framework, this can likely be explained by the fact that the two retrievals estimate different statistics of the a posteriori distribution. The NN-CTTH algorithm has been trained using a squared error loss function which will lead the algorithm to predict the mean of the a posteriori distribution. The QRNN retrieval, on the other hand, predicts the median of the a posteriori distribution. Since the median minimizes the expected absolute error, it is expected that the CTP values predicted by the QRNN yield overall smaller errors.*

**Referee comment:**

2) I understand the usage of the CDF's in figure 3 to demonstrate the statistical value of the retrievals, but again, I think you need to directly show at least an example of a 1-D histogram of the retrieved variable. It gives readers a more direct sense of the variable being retrieved and provides context.

**Author response:**

The following figure and accompanying text have been added to the manuscript:

[Figure]

**Figure 2.** Retrieved a posteriori PDFs corresponding to the CDFs displayed in Figure **??** obtained using MCMC (gray), BMCI (blue), a single QRNN (red line) and an ensemble of QRNNs (red marker).

*Another way of displaying the estimated a posteriori distribution is by means of its probability density function (PDF), which is defined as the derivative of its CDF. From the QRNN output, the PDF is approximated by simply deriving the piece-wise linear approximation to the CDF and setting the boundary values to zero. For BMCI the a posteriori PDF can be approximated a histogram of the CWV values in the database weighted by with the corresponding weights $w_i(\mathbf{y})$. The PDFs for the cases*
5 *corresponding to the CDFs show in Figure 3 are shown in Figure 4.*

**2 Specific comments**

**Referee comment:**

1. Page 2, Line 31: Abbreviations of MAP and 1DVAR are undefined.

**Author response:**

10 The abbreviation MAP has been removed since it neither commonly used nor and accurate designation of the method. The following explanation of the name 1DVAR has been added:

*(also 1DVAR, for one-dimensional variational retrieval)*

**Referee comment:**

2. Page 4, Line 15: The sentence at the beginning of section 2.2 would almost be better suited as a conclusory sentence at the
15 end of section 2.1. As a matter of formatting, I think it's best to avoid a section being a single sentence long.

**Author response:**

The sentence at the beginning of the paragraph was meant as an introduction to the subsection on Bayesian methods. In the hope that this aids both the formatting as well as the readability this introductory part has been changed to:

*Bayesian retrieval methods are methods that use the expression for the a posteriori distribution in Eq. (1) to compute a solution*
20 *to the retrieval problem. Since the a posteriori distribution can generally not be computed or sampled directly, these methods approximate the posterior distribution to varying degrees of accuracy.*

**Referee comment:**

3. Page 9, Line 5: Can you provide an explanation for this restriction of latitudes in the dataset?

**Author response:**

25 We restricted the dataset to the mid-latitudes regime in the hope that this would make the temperature and water vapor profiles more simple to represent using multivariate (Log-)Gaussian distributions. However, since this choice was rather arbitrary and not considered essential for the following analysis, we decided to not explicitly state this in the manuscript.

**Referee comment:**

4. Page 10, Line 8: The formatting of this citation for Typhon seems like it might be incomplete or incorrect for the type of
30 source being cited.

**Author response:**

The AMT guidelines for manuscript preparation do not contain guidelines on how to cite software. The reference format used here seemed most plausible to us and is also the one recommended by the typhon developers. Nonetheless, if the referee has suggestions to improve the citation format, we would be more than happy to consider them.

**Referee comment:**

5. Page 10, Line 19: I don't follow the statement, "not smaller than and larger than 100, respectively". I read that as a logical statement that can't be possible because of the usage of "and." Do you perhaps mean "or?"

**Author response:**

The statement is indeed unnecessarily complicated and probably logically incorrect. It has been rewritten and now reads:

*The retrieval is accepted only if the scale reduction factor is smaller than 1.1 and the effective sample size larger than 100.*

**Referee comment:**

6. Page 10, Line 24: Missing word. Need to insert as into, "It is also released as part of the typhon package.

**Author response:**

The missing word has been inserted.

**Referee comment**

7. Page 15, Line 30: NN-CTTH is an undefined abbreviation. Because of the similarity to the abbreviation for cloud top height (CTH) I almost didn't notice that I didn't understand the name of the algorithm until later in the paper.

**Author response**

We agree with the referee that the introduction of NN-CTTH as name of the retrieval algorithm was not very clear. The relevant phrase has been reformulated and now reads:

*This experiment is based on the work by Håkansson et al. (2018) who developed the NN-CTTH algorithm, a neural network based retrieval of cloud top pressure.*

**Referee comment**

8. Page 17, Line 2: This feels like an incomplete sentence. Perhaps if you reorder it a bit it will make more sense. For example: "The NN-CTTH algorithm and the QRNN are trained using the exact same data set. This training set consists ... "
9. Page 17, Line 7: I also recommend reorganizing this sentence: "In Håkansson et al. (2018) neural networks are trained using different combinations of input features (e.g., [example from paper goes here]) in order to evaluate network performance with different inputs."
10. Page 17, Line 12-14: I don't follow along with this paragraph. Is the "testing under development" dataset the same as the AVHRR version? Could you clarify this para- graph some? Right now it seems that the confusion stems from the discussion of two different datasets within the space of only three sentences – making I difficult to clarify the difference between them. Is the first sentence referring to the first paragraph?

**Author response**

In the hope of making it easier to read and understand, the subsection has been revised and now reads:

*The QRNN uses the same data for training as the reference NN-CTTH algorithm. The data set consists of MODIS Level 1B data (MODIS Characterization Support Team, 2015a, b) collocated with cloud properties obtained from CALIOP (Winker et al., 2009). The top layer pressure variable from the CALIOP data is used as retrieval target. The data was taken from all orbits from 24 days (the 1st and 14th of every month) from the year 2010. In Håkansson et al. (2018) multiple neural networks are*

5 *trained using varying combinations of input features derived from different MODIS channels and ancillary NWP data in order to compare retrieval performance for different inputs. Of the different neural network configurations presented in Håkansson et al. (2018), the version denoted by NN-AVHRR is used for comparison against the QRNN. This version uses only the $11\mu m$ and $12\mu m$ channels from MODIS. In addition to single pixel input, the input features comprise structural information in the form of various statistics computed on a $5 \times 5$ neighborhood around center pixel. The ancillary numerical weather prediction*

10 *data provided to the network consists of surface pressure and temperature, temperatures at five pressure levels, as well as column integrated water vapor. These are also the input features that are used for the training of the QRNN. The training data used for the QRNN are the training and during-training validation set from Håkansson et al. (2018). The comparison to the NN-AVHRR version of the NN-CTTH algorithm uses the dataset for testing under development from Håkansson et al. (2018).*

**Referee comment**

15 11. Page 17, Line 16-17: This sentence is a little confusing to read.

**Author response**

In the hope of making the sentence less confusing it has been rewritten and now reads:
*The training progress, based on which the learning rate is reduced or training aborted, is monitored using the during-training validation dataset from Håkansson et al. (2018).*

20 **Referee comment**

12. Page 18, Line 6-7: Could you expand on how the performance for low and high clouds differs here? Because you're discussing in terms of low and high clouds and cloud top pressure it's useful to highlight that a negative CTPpred-CTPref means that the cloud is higher than the reference. Basically, for the NN-CTTH low clouds have a high-bias and high clouds have a low-bias. It's useful to say that explicitly.

25 **Author response**

This has been addressed as part of the general comment 1.).

**Referee comment**

13. Page 23, Line 20: For the sake of clarity for a more general audience can you provide an example of a vector-valued retrieval quantity.

30 **Author response**

The sentence has been rewritten and now includes an example of a vector-valued retrieval quantity:

*Another aspect of the application of QRNNs to remote sensing retrievals that remains to be investigated is how they can be used to retrieve vector-valued retrieval quantities, such as concentration profiles of atmospheric gases or particles.*